# Albuminuria Is Affected by Urinary Tract Infection: A Comparison between Biochemical Quantitative Method and Automatic Urine Chemistry Analyzer UC-3500

**DOI:** 10.3390/diagnostics13213366

**Published:** 2023-11-02

**Authors:** Yi-Ju Chou, Chun-Chun Yang, Shang-Jen Chang, Stephen Shei-Dei Yang

**Affiliations:** 1Division of Urology, Taipei Tzu Chi Hospital, Buddhist Tzu Chi Medical Foundation, New Taipei 23142, Taiwan; andy0611andy@gmail.com; 2Department of General Laboratory, Taipei Tzu Chi Hospital, Buddhist Tzu Chi Medical Foundation, New Taipei 23142, Taiwan; tch34066@tzuchi.com.tw; 3Department of Urology, National Taiwan University Hospital, Taipei 10002, Taiwan; krissygnet@gmail.com; 4School of Medicine, Tzu Chi University, Hualien 97004, Taiwan

**Keywords:** albuminuria, diagnosis, automation laboratory, urinary tract infection

## Abstract

The automated urine reagent strip test is a cost-effective tool for detecting albuminuria in patients. However, prior research has not investigated how urinary tract infections (UTIs) affect the test’s accuracy. Therefore, this study aims to assess the impact of UTIs on albuminuria diagnosis using both the biochemical quantitative method and the test strip method of the Fully Automatic Urine Chemistry Analyzer, UC-3500 (Sysmex, Kobe, Japan). From March to December 2019, we prospectively collected midstream urine from adult female UTI patients before and after one week of cephalexin treatment. The urine samples were subjected to culture, routine urinalysis, and albuminuria diagnosis using the biochemical quantitative method and UC-3500. Albuminuria was defined as a urine albumin to creatinine ratio (UACR) ≥ 30 mg/g in the biochemical quantitative method. The results were compared between the two methods. Among fifty-four female patients (average age: 50.5 ± 4.4 years) with UTIs, 24 (44.44%) had transient albuminuria. The quantitative UACR significantly decreased after one week of antibiotic treatment (median: 53 mg/g to 9 mg/g; median difference: −0.54, *p* < 0.0001). UC-3500 exhibited a higher false positive rate for diagnosing albuminuria during UTIs (42%) compared to after treatment (19%). Its agreement with the biochemical quantitative method was moderate during UTI (κ = 0.49, 95% confidence interval [CI]: 0.24–0.73) and good after treatment (κ = 0.65, 95% CI: 0.45–0.86). UC-3500’s accuracy in diagnosing albuminuria is influenced by UTIs, leading to either transient albuminuria or a false positive reaction of the test strip. UTI should be excluded or treated before its application in albuminuria screening.

## 1. Introduction

Chronic kidney disease (CKD) commonly results from hypertension and diabetes mellitus (DM), with albuminuria serving as an indicator of kidney damage. Timely detection of albuminuria is essential for clinicians to provide early treatment and potentially prevent patients from requiring hemodialysis [1]. The Kidney Disease: Improving Global Outcomes (KDIGO) 2012 guideline recommends identifying albuminuria using the urine albumin to creatinine ratio (UACR), defining a value of 30 mg/g or more (3 mg/mmol) as the cut-off point [1]. While the biochemical quantitative method serves as the gold standard for measuring UACR, it is both costlier and more time-consuming compared to reagent strip tests [2]. Consequently, previous studies have explored reagent strip tests with automatic reading functions as an alternative for diagnosing albuminuria, and they demonstrate promising results [3,4,5].

One of these automated urine reagent strip analyzers is UC-3500 (Sysmex, Kobe, Japan). It employs reflectance photometry with a color complementary metal oxide semiconductor detector (CMOS) to interpret color changes on the reagent strip, resulting in objective results that are less susceptible to human bias. The UC-11A MEDITAPE test strip measures 11 urine parameters, including protein, albumin, and creatinine, and provides semi-quantitative readings for urine protein to creatinine ratio (UPCR) and UACR. The semi-quantitative UACR results (0: <30 mg/g, 1: 30–300 mg/g, 2: >300 mg/g) correspond to the albuminuria categories defined by the KDIGO guideline (A1: <30 mg/g, A2: 30–300 mg/g, A3: >300 mg/g), facilitating the evaluation of albuminuria severity [1]. Previous studies have reported good accuracy of UC-3500 compared to the biochemical quantitative method [6,7].

UC-3500, as outlined in its user manual, utilizes the protein error of pH indicator method to detect urine protein and albumin [8]. Both pads contain tetrabromophenol blue as the dye for the reaction. However, this dye does not react specifically to urine albumin and can generate false positive results in the presence of detergent, hematuria, immunoglobulin, or other urine proteins. It is important to note that approximately 63–80% of urinary tract infections (UTI) are associated with proteinuria, and 68–91.7% are accompanied by hematuria [9,10,11]. Therefore, the protein and albumin pads may produce false positive results in patients with UTIs.

While previous research has consistently demonstrated that UTIs can lead to a transient increase in urinary albumin, there has been limited exploration of how UTIs may impact the accuracy of reagent strip tests [12,13,14]. The dye used in UC-3500 to detect albumin also reacts with other substances in urine, potentially making it more susceptible to the influence of UTIs. Although this test is convenient for screening albuminuria, result interpretation requires caution. This study aims to investigate whether UTIs affect the accuracy of UC-3500, a reagent strip test, in diagnosing albuminuria by comparing the results with the gold standard method for diagnosing albuminuria, the biochemical quantitative method.

## 2. Materials and Methods

### 2.1. Subjects

Between March and December 2019, we enrolled female patients visiting our urologic outpatient department with symptoms indicative of UTIs. Participants completed a baseline characteristics questionnaire and a urinary tract infection symptom assessment questionnaire (UTISA). Those with UTISA scores exceeding 3 were deemed to have UTIs [15] and were included in the study. Treatment consisted of oral cephalexin 500 mg four times daily for seven days due to its recognized efficacy as a first-line antibiotic against common uropathogens, based on reports from Taiwan [16,17]. Exclusion criteria encompassed patients with a fever exceeding 38 °C, pregnancy, genitourinary tract anomalies, bladder cancer, indwelled urethral catheters due to urinary retention, chronic kidney disease with dialysis, and recent antibiotic use within one week.

### 2.2. Urine Sampling

Random urine samples were collected during the initial visit and a follow-up visit after one week of antibiotic treatment. Prior to urine sampling, a nurse instructed patients to provide a clean catch mid-stream urine specimen to minimize contamination. Within 30 minutes of collection, samples were sent for urinalysis using the UC-3500 and Fully Automated Urine Particle Analyzer UF-1000i by Sysmex, Kobe, Japan, urine culture, and quantitative analysis of UPCR and UACR. Results for urine leukocyte esterase and nitrite were obtained using UC-3500, while urine white blood cell (WBC), red blood cell (RBC), and bacteria concentration were obtained using UF-1000i.

### 2.3. Biochemical Quantitative Method

The quantitative measurement of urine protein, albumin, and creatinine was performed using the Dimension^®^ Clinical Chemistry System (Dade Behring, Newark, NJ, USA). This method assesses light absorbance at specific wavelengths to determine the analyte’s concentration. The higher the quantity, the greater the absorbance. Urine protein is determined by the 540 nm light absorbance of the product from the Biuret reaction (measurement range: 2–12 g/dL). Urine albumin is quantified by the 600 nm light absorbance of the albumin–bromocresol purple complex (measurement range: 0.6–8 g/dL). Urine creatinine is assessed through the 510 nm light absorbance of the product from the Jaffe reaction (measurement range: 0–20 mg/dL).

### 2.4. The Influence of Albuminuria Caused by UTI

The study population was classified into three groups according to changes in UACR before and after antibiotic treatment. Negative albuminuria was defined as having a UACR of less than 30 mg/g both before and after antibiotic treatment. Transient albuminuria was defined as a decrease in quantitative UACR from over 30 mg/g before antibiotic treatment to less than 30 mg/g after treatment. Persistent albuminuria was defined as a quantitative UACR exceeding 30 mg/g both before and after treatment. To comprehend how UTIs lead to albuminuria, differences in the UTISA questionnaire score and other urine parameters between the three albuminuria groups were compared.

### 2.5. UC-3500

UC-3500 interprets the test strip through a CMOS sensor, measuring the light reflected from the reagent strip pad [6]. A decrease in reflected light corresponds to a higher analyte quantity. The protein pad contains 0.015 mg of tetrabromophenol blue, while the albumin pad contains 0.01 mg of tetrabromophenol blue. Both detect urine protein (measurement range: 15–1000 mg/dL) and albumin (measurement range: 10–150 mg/L) using the protein error of the pH indicator method. The creatinine pad contains 0.48 mg of 3,5-dinitrobenzoic acid and generates a garnet-red color in the presence of creatinine (measurement range: 10–300 mg/dL).

### 2.6. The Influence of UTI on the Accuracy of UC-3500

Albuminuria was identified by a positive UACR using UC-3500 or a quantitative UACR ≥ 30 mg/g using the biochemical quantitative method. The severity of albuminuria was assessed using the albuminuria categories from the KDIGO guideline. The agreement between UC-3500 and the biochemical quantitative method in diagnosing the presence and severity of albuminuria was compared. The impact of UTI on the diagnostic accuracy of UC-3500 was evaluated by calculating the differences in agreement between the two methods before and after antibiotic treatment. As the albumin pad of the UC-3500 reagent strip may generate false positive results in the presence of urine substances other than albumin [8], differences in other urine parameters between cases with false positive results and cases with true negative results were also compared to identify potential confounders interfering with the albumin pad.

### 2.7. Statistical Analysis

All data were analyzed using MedCalc Statistical Software version 19.1.3 (MedCalc Software bv. Ostend, Belgium; https://www.medcalc.org; accessed on 23 October 2023). The Shapiro–Wilk test was employed to determine the normality of continuous data. Parametric data were reported as mean ± standard deviation (SD), while non-parametric data were presented as median and interquartile range (IQR). Cohen’s kappa [18] and Spearman rank correlation analysis were used to assess the agreement between UC-3500 and the biochemical quantitative method in terms of the diagnosis and severity of albuminuria. The degree of agreement was classified according to Altman’s classification [19] (<0.2: poor, 0.21–0.4: fair, 0.41–0.6: moderate, 0.61–0.8: good, >0.8: very good). The accuracy of UC-3500 in diagnosing proteinuria was assessed through receiver operating characteristic curve (ROC), and was expressed in terms of sensitivity, specificity, and area under the curve (AUC). The difference in independent dichotomous data (nitrite) was compared using the Chi-squared test. The difference in independent ordinal data (leukocyte esterase) and continuous data (WBC, RBC, and bacteria) was analyzed via the Mann–Whitney test, while paired data were compared through the Wilcoxon signed-rank test. Multiple comparisons were conducted using the Kruskal–Wallis test. A *p*-value < 0.05 was considered statistically significant.

## 3. Results

### 3.1. Patient Characteristics

A total of 113 patients were initially enrolled in the study. However, 58 patients were excluded due to their inability to provide UACR results using UC-3500, primarily due to diluted urine. One patient was further excluded because of inadequate sample availability, preventing the quantification of UACR. Consequently, 54 female patients, with a mean age of 50.5 ± 4.4 years, were included for analysis. Table 1 presents the essential baseline characteristics of the study subjects. Among these, seven patients (12.96%) had DM, and ten patients (18.52%) had hypertension. The median total score of the UTISA questionnaire before treatment was 10.5 (IQR: 8–13), and there was a significant decrease in the total score after treatment (median: 1, IQR: 0–4, *p* < 0.001). Following antibiotic treatment, the UTISA questionnaire scores were similar among the three albuminuria groups (Appendix A). The results of urine bacterial cultures showed that the majority of bacteria were Gram-negative, accounting for 62.96% of the total. *E. coli* was the most frequently cultured bacterium, representing 51.85% of the total.

### 3.2. The Influence of Antibiotics Treatment on Results of Quantitative UACR

Out of the 35 (64.8%) patients with quantitative UACR ≥ 30 mg/g before treatment, 11 (27.5%) exhibited persistent albuminuria, while 24 (44.4%) displayed transient albuminuria. Among those with persistent albuminuria, one patient had hypertension, one had DM, one had both hypertension and DM, and the remaining eight reported no chronic illnesses. Both quantitative UPCR (median: 232 mg/g to 125 mg/g; median difference: −140, *p* < 0.001) and quantitative UACR (median: 53 mg/g to 9 mg/g; median difference: −0.54, *p* < 0.001) exhibited significant decreases after treatment. There were notable decreases in quantitative UACR within the persistent, transient, and negative albuminuria subgroups (Figure 1).

### 3.3. Differences among the Three Albuminuria Groups

Table 2 illustrates the comparison of various variables among the three albuminuria groups. Prior to treatment, the UTISA questionnaire scores were similar across all three albuminuria groups. However, significant differences were observed among the groups regarding nitrite presence, leukocyte esterase, WBC count, and RBC count. Both the persistent albuminuria and transient albuminuria groups exhibited a higher proportion of positive nitrite compared to the negative albuminuria group. Additionally, the transient albuminuria group had higher levels of leukocyte esterase, WBC count, and RBC count compared to the negative albuminuria group. There were no significant differences among the three groups in terms of urine bacterial quantity.

### 3.4. The Influence of UTI on the Accuracy of UC-3500

The semi-quantitative results from UC-3500 demonstrated a significant correlation with the biochemical quantitative UACR both before (Spearman’s rho = 0.68, *p* < 0.001) and after (Spearman’s rho = 0.69, *p* < 0.001) antibiotic treatment. Before antibiotic treatment, the agreement between UC-3500 and the biochemical quantitative method for the diagnosis of albuminuria was moderate (κ = 0.49, 95% confidence interval (CI): 0.24–0.73). The sensitivity and specificity of UC-3500 for diagnosing albuminuria were 88.57% and 57.89%, respectively (Table 3). Post-treatment agreement was good (κ = 0.65, 95% CI: 0.45–0.86). After treatment, the sensitivity and specificity of UC-3500 increased to 100.00% and 80.95%, respectively (Table 4). There was improved agreement in the severity of albuminuria after treatment compared to before treatment (Before: κ = 0.47, 95% CI: 0.28–0.65; After: κ = 0.65, 95% CI: 0.46–0.84, Table 5 and Table 6). The accuracy of UC-3500 in diagnosing albuminuria also improves after UTI treatment. The pre-treatment AUC was 0.76 (95% CI: 0.62–0.87), while the post-treatment AUC increased to 0.93 (95% CI: 0.83–0.98) (Figure 2 and Figure 3). During infection, UC-3500 exhibited higher accuracy in diagnosing severe albuminuria (A1 = 57.9%, A2 = 53.8%, A3 = 88.9%). Of the 19 patients with quantitative UACR < 30 mg/g before treatment, UC-3500 detected eight positive UACR results (false positive) and eleven negative UACR results (true negative). All 19 patients had negative nitrite. UPCR, leukocyte esterase, and WBC were not significantly different between those with false positive results and those with true negative results. The false positive group had a significantly higher concentration of RBC and bacteria in their urine (Table 7).

## 4. Discussion

According to the findings of this study, UTI can influence the diagnosis of albuminuria in two significant ways: (1) the emergence of transient albuminuria that resolves after antibiotic treatment and (2) the occurrence of false positive results in reagent strip tests, such as the UC-3500. UACR serves as a crucial marker for kidney damage in conditions like chronic kidney disease, DM, and hypertension [1,20,21]. As a result, numerous studies have explored the potential use of reagent strip tests as an alternative diagnostic tool for early detection [3,4,5]. However, despite the promising prospects of reagent strip tests, no study, to our knowledge, has ventured into examining the possible impact of UTI on albuminuria diagnosis. Remarkably, even the UC-3500 user manual does not caution that UTI may lead to false positive results on the albumin pad. This study stands as the pioneering investigation to unveil that UTI not only leads to transient albuminuria, but also casts a shadow of doubt on the accuracy of reagent strip tests for albuminuria diagnosis. These findings underscore the necessity for vigilance when employing this convenient screening method for albuminuria, as it is evidently more susceptible to UTI’s influence compared to the biochemical quantitative method. Therefore, the presence of UTI must be rigorously excluded before embarking on the screening process.

Previous studies examining the relationship between UTI and albuminuria have consistently pointed towards a robust association between albuminuria and symptomatic UTI. Additionally, following antibiotic treatment, albuminuria generally shows improvement. In contrast, asymptomatic bacteriuria appears unrelated to albuminuria [13]. In the present study, we specifically enrolled patients with a UTISA questionnaire score > 3, designating them as individuals with symptomatic UTI. Consequently, our results align with this pattern, revealing a significant reduction in quantitative UACR following antibiotic treatment. Moreover, out of the 35 patients who exhibited albuminuria prior to treatment, 24 (68.5%) witnessed a return to normal urinary albumin levels after treatment, thus conforming to the definition of transient albuminuria.

This substantial reduction was not confined solely to the transient albuminuria group but extended to the persistent and negative albuminuria subgroups (Figure 1). This implies that UTI can indeed augment urinary albumin excretion in all patients, albeit to varying degrees. Certain patients may even reach the threshold of albuminuria as defined by quantitative UACR. Significantly, the transient albuminuria group exhibited higher levels of leukocyte esterase, WBC, and RBC when compared to the negative albuminuria group (Table 2). It is plausible that these patients may have endured more pronounced inflammation in the urinary bladder, which could have led to concurrent hematuria or the release of exudate from blood vessels, thereby increasing the amount of albumin in the urine [22,23]. In contrast, the persistent albuminuria group showed no significant differences in WBC and RBC compared to the negative albuminuria group. This indicates that the presence of albumin in the urine in this group is not solely due to infection-related inflammation but may be indicative of an underlying condition of albuminuria. Among this group, three patients had known chronic conditions associated with albuminuria, while the remaining eight patients may require further investigation to identify the underlying causes of their albuminuria.

Despite the good correlation between the semi-quantitative results of UACR obtained from UC-3500 and the biochemical quantitative UACR, the agreement between UC-3500 and the biochemical quantitative method was only moderate (κ = 0.49) for diagnosing albuminuria in UTI patients. Nevertheless, this agreement improved to a good level after antibiotic treatment (κ = 0.65). Prior to treatment, the sensitivity and specificity were measured at 88.57% and 57.89%, respectively (Table 3). Following treatment, the sensitivity and specificity rose to 100.00% and 80.95%, aligning with the results of a previous study that assessed the performance of UC-3500 in a non-UTI population [2]. The AUC of UC-3500 for diagnosing albuminuria also increased after antibiotic treatment (Figure 2 and Figure 3). The augmentation in agreement for classifying albuminuria severity between the two methods was also evident after infection control. These findings underscore the importance of effectively treating the infection to secure reliable results when employing UC-3500 for albuminuria classification.

Furthermore, we observed a heightened false positive rate of UC-3500 for albuminuria diagnosis during UTI. This inaccuracy may stem from the varying specificity to albumin of the dyes used by UC-3500 and the biochemical quantitative method. Bromocresol purple used in the biochemical quantitative method reacts specifically with albumin [24,25], in contrast to tetrabromophenol blue used in UC-3500 [26]. Urine RBC and bacteria concentration were significantly higher in the false positive group (Table 7), implying their interference with the albumin pad. Hematuria is already acknowledged to cause a false positive reaction [8], while a high bacteria concentration may elicit a stronger immune response and higher production of immunoglobulin. Nonetheless, the insignificant difference in proteinuria between the false positive and true negative groups contrasts with our expectation. It can be explained by the fact that albumin comprises the primary content of proteinuria during UTI. The comparable incidence of albuminuria (64.82%) in this study to the incidence of proteinuria found in early studies further supports this explanation [9,10,11].

Patients with DM require regular monitoring for the presence of albuminuria or to track changes in its severity. However, DM patients are more susceptible to UTI than the general population [27]. Consequently, using a reagent strip test for screening or monitoring in this patient group may render the results more susceptible to the influence of UTI. Nevertheless, the test strip of UC-3500 can simultaneously measure leukocyte esterase and nitrite. Furthermore, it can be used in conjunction with an automatic urine particle analyzer to determine the presence of UTI [28]. This additional information in the test results of these patients can help us assess whether the UACR measured by UC-3500 might be affected by the concomitant presence of UTI. This is a critical factor that makes UC-3500 a more suitable tool for albuminuria screening than the biochemical quantitative method.

This study is not without limitations. Firstly, it constitutes a secondary analysis of data from our prior research [29], which means we did not assess the within-run and between-run variations of UACR measurement. Secondly, the exclusion of numerous patients due to the UC-3500’s inability to analyze UACR in diluted urine resulted in a relatively limited sample size. This also underscores the challenges in employing UC-3500 for diagnosing albuminuria during UTI, as most UTI patients tend to increase their fluid intake, which can lead to urine dilution. Finally, we utilized post-treatment examination results as the control group to evaluate whether the accuracy of albuminuria diagnosis was indeed affected by the presence of infection. Opting for individuals without UTIs as the control group might have been more appropriate. However, the retrospective nature of this study precluded the use of patients without UTIs as a control group. Nonetheless, the significant improvement in post-treatment UTISA questionnaire scores and the insignificant difference of post-treatment UTISA questionnaire score between the three albuminuria groups suggest that patients with persistent albuminuria may indeed have genuine albuminuria rather than untreated infections. A future study, featuring a larger participant pool and a prospective design, is warranted to further validate the results obtained in this study.

## 5. Conclusions

UTI have a notable impact on the detection of albuminuria when using UC-3500. This influence can manifest in two significant ways, either by causing a false positive reaction on the albumin pad or by inducing a transient albuminuria. As such, it is essential to acknowledge UTI as a significant confounding factor. Prior to employing the cost-effective UC-3500 as a diagnostic tool for detecting albuminuria and evaluating its severity, it is imperative to exclude UTIs or ensure their effective treatment.

## Figures and Tables

**Figure 1 diagnostics-13-03366-f001:**
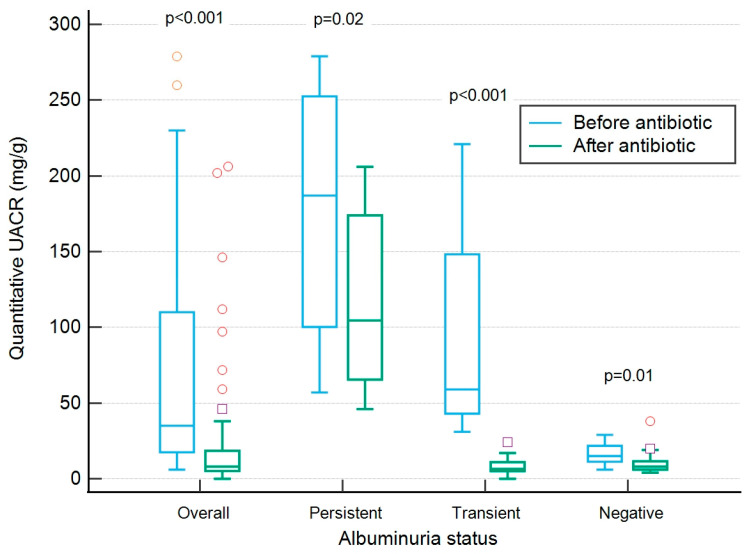
Results of quantitative UACR before and after antibiotics treatment.

**Figure 2 diagnostics-13-03366-f002:**
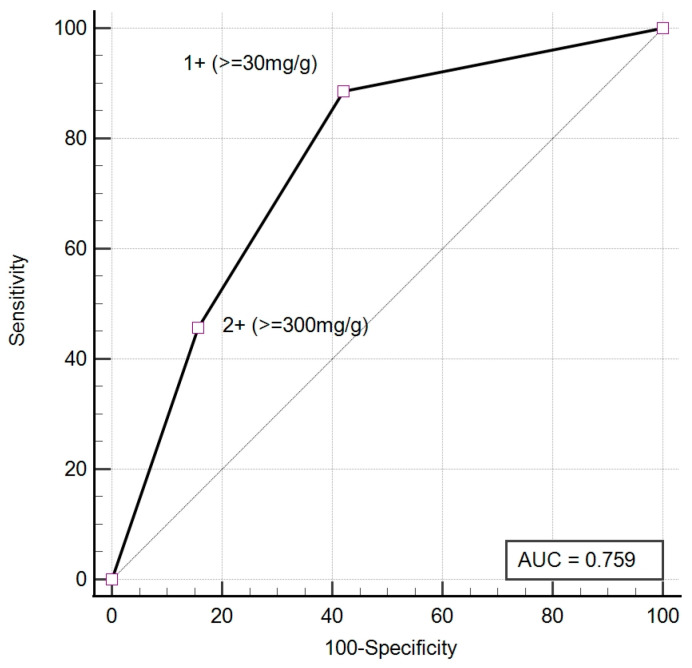
The ROC of UC-3500 for diagnosing albuminuria before treatment.

**Figure 3 diagnostics-13-03366-f003:**
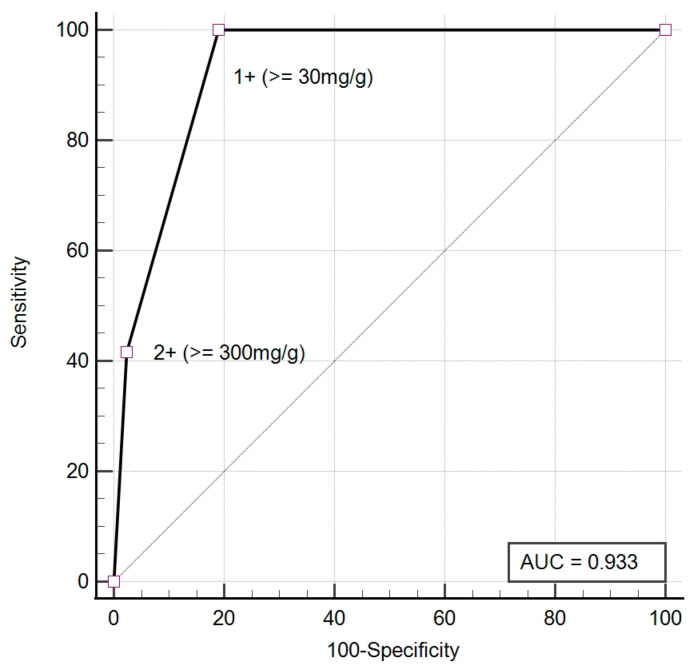
The ROC of UC-3500 for diagnosing albuminuria after treatment.

**Table 1 diagnostics-13-03366-t001:** Baseline characteristics of the study subjects.

Baseline Characteristics	
Underlying condition, *N* (%)	
Hypertension	10 (18.52%)
Diabetes mellitus	7 (12.96%)
Culture pathogen, *N* (%)	
Gram-positive	8 (14.82%)
Gram negative	34 (62.96%)
Mixed growth	12 (22.22%)
Total score of UTISA, median (IQR)	10.5 (8–13)
UACR ≥ 30 mg/g, *N* (%)	35 (64.8%)

**Table 2 diagnostics-13-03366-t002:** Comparison of UTISA score and urine parameters between different albuminuria groups before antibiotics treatment.

	Negative Albuminuria (*n* = 19)	Transient Albuminuria (*n* = 24)	Persistent Albuminuria (*n* = 11)	*p*-Value
UTISA score ^#^	11.0 (8.3–13.0)	11.0 (8.5–14)	10 (6.3–12.8)	0.716
Nitrite (+) *	0 (0%) ^†,‡^	9 (37.5%) ^†^	5 (45.5%) ^‡^	0.005
Leukocyte esterase ^#^	1 (0–3) ^†^	3 (2–3) ^†^	2 (0.2–3)	0.032
WBC (/µL) ^#^	75.5 (50.1–532.3) ^†^	787.1 (453.1–2402.1) ^†^	683.0 (76.6–1682.0)	0.003
RBC (/µL) ^#^	19.9 (7.9–31.8) ^†^	108.5 (43.2–517.5) ^†^	40.5 (15.9–148.0)	0.001
Bacteria (/µL) ^#^	487.1 (102.8–1528.1)	803.8 (226.6–11,574.0)	2348.7 (109.7–11,677.7)	0.292

* Expressed as proportion; ^#^ expressed as median and interquartile range; ^†^ significance between negative and transient albuminuria groups; ^‡^ significance between negative and persistent albuminuria groups.

**Table 3 diagnostics-13-03366-t003:** Pre-treatment agreement * for diagnosis of albuminuria between UC-3500 and biochemical quantitative method.

	UC-3500 (+)	UC-3500 (−)	Total
Quantitative method (+)	31	4	35
Quantitative method (−)	8	11	19
Total	39	15	54

* Kappa value = 0.49, moderate agreement.

**Table 4 diagnostics-13-03366-t004:** Post-treatment agreement * for diagnosis of albuminuria between UC-3500 and biochemical quantitative method.

	UC-3500 (+)	UC-3500 (−)	Total
Quantitative method (+)	12	0	12
Quantitative method (−)	8	34	42
Total	20	34	54

* Kappa value = 0.65, good agreement.

**Table 5 diagnostics-13-03366-t005:** Pre-treatment agreement * for the severity of albuminuria between UC-3500 and biochemical quantitative method.

		UC-3500			Total
		A1	A2	A3	
Quantitative method	A1	11	5	3	19
	A2	4	14	8	26
	A3	0	1	8	9
Total		15	20	19	54

* Kappa value = 0.47, moderate agreement.

**Table 6 diagnostics-13-03366-t006:** Post-treatment agreement * for the severity of albuminuria between UC-3500 and biochemical quantitative method.

		UC-3500			Total
		A1	A2	A3	
Quantitative method	A1	34	7	1	42
	A2	0	7	2	9
	A3	0	0	3	3
Total		34	14	6	54

* Kappa value = 0.65, good agreement.

**Table 7 diagnostics-13-03366-t007:** Difference of urine parameters between false positive and true negative groups before treatment.

	False Positive Group (*n* = 8)	True Negative Group (*n* = 11)	*p*-Value
Leukocyte esterase *	2.5 (1–3)	0 (0–2)	0.063
UPCR (mg/g) *	156 (122.5–190)	119 (92–152)	0.107
RBC * (/µL)	31.6 (25.3–76.3)	14.1 (5.9–18.8)	0.007
WBC * (/µL)	368.1 (111.6–832.5)	60.2 (11.2–256.9)	0.099
Bacteria * (/µL)	1372.7 (911.8–5184.2)	110.7 (37.3–355.6)	<0.001

* Expressed as median and interquartile range.

## Data Availability

The data presented in this study are available on request from the corresponding author. The data are not publicly available due to privacy issue.

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
