# Peer review of "Albuminuria Is Affected by Urinary Tract Infection: A Comparison between Biochemical Quantitative Method and Automatic Urine Chemistry Analyzer UC-3500"

_diagnostics, 2023, doi:10.3390/diagnostics13213366_

Round 1

Reviewer 1 Report

Comments and Suggestions for Authors

Abstract: 

1. No need to include the words “background”, “methods”, etc. the abstract states the objective but doesn’t include an actual background sentence.  

2. Include an initial background sentence.

Introduction: 

1. Needs a reference: The 48 semi-quantitative results of UACR (0: 300mg/g) correspond 49 to the definitions of albuminuria categories in the KDIGO guideline (A1: 300mg/g), which facilitates the evaluation of the severity of albuminuria.  

Results:  

1. How do the 11 cases of persistent albuminuria overlap with the 7 cases of diabetes and the 10 cases of hypertension? Did any patients without these complications have persistent albuminuria?

2. It would be nice to see a table with the results of the UTISA symptom survey.

Methods: OK

Discussion: 

1. We appreciate that the authors discussed the sample size limitation in the Discussion. It was surprising that “58 patients were 143 excluded due to the inability to obtain UACR results using UC3500”. We would encourage the authors to consider a future study which either limits this high exclusion rate, or enrolls additional participants to create a larger final sample size. We further encourage the authors to consider including male participants at that time.

Comments on the Quality of English Language

Author Response

Please see the attachment. Thank you !

Reviewer 2 Report

Comments and Suggestions for Authors

The paper entitled "Albuminuria Is Affected by Urinary Tract Infection: A Comparison Between Biochemical Quantitative Method and Automatic Urine Chemistry Analyzer UC-3500"is carefully read and reviewed. Authors studied the proteinuria in patients with urinary tract infection. The study has some novelty but also have some issues to overcome.

Transient proteinuria is a well established issue in patients with urinary tract infections (Haider MZ, Aslam A. Proteinuria. In: StatPearls. StatPearls Publishing, Treasure Island (FL); 2022. PMID: 33232060). Thus, novelty of the study is questionable. However, studying the proteinuria by a specific device has some novelty. Please improve introduction.

Another issue is the antibiotic that patients were prescribed to use. Were all microorganisms that detected in urinary culture sensitive to Cephalexin? Please clarify.

Methodology is very clear and free from flaws. Results were also expressed well. However, thick digit expression of the significant p values is recommended. Moreover, I suggest giving three decimals in p values and two decimals in other values. Additionally, in table six, authors noted that the data was expressed as medians and interquartile ranges. However, it looks like as medians and (min-max). 

 Discussion can be improved. Discuss the results in line with existing literature data please. For example, there are conflicting results about proteinuria associated with urinary infections. While febrile urinary tract infections cause proteinuria, asymptomatic cases were not associated with increased protein excretion (Joanne L. Carter, Charles R. V. Tomson, Paul E. Stevens, Edmund J. Lamb, Does urinary tract infection cause proteinuria or microalbuminuria? A systematic review, Nephrology Dialysis Transplantation 2006;21(11):3031–3037. https://doi.org/10.1093/ndt/gfl373). Moreover, authors are advised to comment on the clinical significance of the study results in discussion. 

Author Response

Please see the attachment. Thank you !

Reviewer 3 Report

Comments and Suggestions for Authors

In this paper, Yi-Ju Chou and colleagues validated the UC-3500 urinalysis system for detecting proteinuria in urinary tract-infected individuals. There is a limited number of publications related to the validation of different laboratory equipment; thus, the importance of the given research must be highlighted. Unfortunately, some flows in the study should be improved for clarity.

Major:

1. The study lacks a healthy control group, and the affected population was divided according to the tested parameter (proteinuria) for further comparisons. Also, some patient characteristics tables would be valuable.

2. The information on urine sampling is limited. Was there any desired timing for urine sampling, like the first-morning urine? It was a significant clue for the study as 58/113 samples were excluded due to “diluted urine” (page 4, line 143). Was this phenomenon covered by inclusion/exclusion criteria in the study?

Minor:

1. Please give an analytical range for protein detection in all methods used in the study (LOD/LOQ).

2. Please cover the laboratory guidelines (methods) for proteinuria testing in the discussion section. How does it, if any, correspond with your results?

3. Please add a figure for the results in the 3.3 section (sensitivity and specificity). Figure 1 needs reformatting – please add some clusters to the y-axis to make the bars bigger.

4. Please consider citing:

doi: 10.3390/ijerph17124195

doi: 10.1080/00325481.2019.167954

Author Response

Please see the attachment. Thank you !

Round 2

Reviewer 2 Report

Comments and Suggestions for Authors

Revisions done in the text are satisfactory. I recommend publication  of the paper in its current form.

Reviewer 3 Report

Comments and Suggestions for Authors

Authors have improved paper accordingly. I have no more questions.